# Epigenetic Regulation of MAP3K8 in EBV-Associated Gastric Carcinoma

**DOI:** 10.3390/ijms24031964

**Published:** 2023-01-19

**Authors:** Gaurab Roy, Ting Yang, Shangxin Liu, Yi-Ling Luo, Yuantao Liu, Qian Zhong

**Affiliations:** State Key Laboratory of Oncology in South China, Collaborative Innovation Center for Cancer Medicine, Sun Yat-Sen University Cancer Center, Guangzhou 510060, China

**Keywords:** epigenetics, mitogen-activated protein kinase 8 (*MAP3K8*), Epstein–Barr virus-associated gastric carcinoma (EBVaGC), super-enhancer (SE), CRISPR interference (CRISPRi)

## Abstract

Super-enhancers (SEs) regulate gene expressions, which are critical for cell type-identity and tumorigenesis. Although genome wide H3K27ac profiling have revealed the presence of SE-associated genes in gastric cancer (GC), their roles remain unclear. In this study, ChIP-seq and HiChIP-seq experiments revealed mitogen-activated protein kinase 8 (*MAP3K8*) to be an SE-associated gene with chromosome interactions in Epstein–Barr virus-associated gastric carcinoma (EBVaGC) cells. CRISPRi mediated repression of the *MAP3K8* SEs attenuated *MAP3K8* expression and EBVaGC cell proliferation. The results were validated by treating EBVaGC cells with bromodomain and the extra-terminal motif (BET) inhibitor, OTX015. Further, functional analysis of *MAP3K8* in EBVaGC revealed that silencing *MAP3K8* could inhibit the cell proliferation, colony formation, and migration of EBVaGC cells. RNA-seq and pathway analysis indicated that knocking down *MAP3K8* obstructed the notch signaling pathway and epithelial-mesenchymal transition (EMT) in EBVaGC cells. Further, analysis of the cancer genome atlas (TCGA) and GSE51575 databases exhibited augmented *MAP3K8* expression in gastric cancer and it was found to be inversely correlated with the disease-free progression of GC. Moreover, Spearman’s correlation revealed that *MAP3K8* expression was positively correlated with the expressions of notch pathway and EMT related genes, such as, *Notch1, Notch2,* C-terminal binding protein 2 (*CTBP2*), alpha smooth muscle actin isotype 2 (*ACTA2*), transforming growth factor beta receptor 1 (*TGFβR1*), and snail family transcriptional repressors 1/2 (*SNAI1*/*SNAI2*) in GC. Taken together, we are the first to functionally interrogate the mechanism of SE-mediated regulation of *MAP3K8* in EBVaGC cell lines.

## 1. Introduction

Gastric cancer (GC) is the fourth largest cause of cancer-related deaths worldwide [1] with a five-year relative survival of 20–30% in most areas of the world [2]. Epstein–Barr virus-associated gastric carcinoma (EBVaGC) is linked with Epstein–Barr virus (EBV) infection and is one of the four GC subtypes defined by the cancer genome atlas (TCGA) [3]. EBVaGC possesses numerous discrete genomic or epigenomic characteristics and clinicopathological attributes compared with other molecular subtypes of GC [4]. A meta-analysis of EBV prevalence in GC patients revealed that EBV infection could increase GC risk by more than 18 times [5]. EBV infection silences major tumor suppressor genes, cell cycle genes, and cellular differentiation factors via hypermethylation and nurtures a highly proliferative and poorly differentiated cell population [6]. Although around 10% of gastric cancers worldwide are associated with EBV infection [7], concrete evidence of EBVs distinct etiological role in GC development is still lacking [8].

Of late, cancer epigenetics has emerged as a fundamental oncogenic mechanism leading to the establishment of hallmark cancer cell behaviors. Epigenetic changes cooperate through a network of mutually reinforcing or counteracting signals and play key roles in the dysregulation of oncogenes and tumor suppressor genes. Epigenetic regulation of cancer-associated genes has been implicated in the progression and therapy response of several cancers including, colorectal cancer, breast cancer, gastrointestinal cancer, etc. [9,10,11]. Moreover, drugs targeting the epigenetic markers are extensively studied to open new avenues for cancer treatment [12,13]. Super-enhancers (SEs) have emerged as master gene regulators and a novel frontier for the epigenetic regulation of carcinogenesis [14,15]. Super-enhancer-driven transcriptional programs lead to aberrant activation of oncogenic and tumor suppressive pathways [16]. A recent report revealed that super-enhancer-mediated transcriptional regulation of *TGFβR2* signaling was responsible for the aggression and metastasis of pancreatic cancer [17]. Similarly, a super-enhancer-regulated gene, *IL-20RA*, controlled the growth and metastasis of colorectal cancer by modulating oncogenic and immune pathways [18]. In hepatocellular carcinoma (HCC), transcriptional factor 4 (TCF4) promoted *AJUBA* transcription and thereby HCC progression by binding to its SE [19]. Likewise, a SE controlled the expression of *RCAN1.4*, a potential tumor suppressor in breast cancer [20]. Although super-enhancer-associated genes have been reported in gastric carcinoma, their function is yet to be validated [21]. Therefore, based on the above studies and the caveat surrounding the role of SEs in EBVaGC progression, a robust functional analysis of the SE-associated gene/s in EBVaGC must be performed.

Over the last decade, novel perceptions have been obtained into the mechanisms by which regulatory elements, such as SEs, etc., can act over large genomic distances [22,23]. A substantial number of investigations have highlighted the role of both intrachromosomal and interchromosomal interactions for recurrent physiological gene-expression programs [24,25]. H3K27ac HiChIP analyses identified interferon regulatory factor 9 (*IRF9*) as a major direct target of stromal antigen 2 (*STAG2*) in melanoma cells [26]. Similarly, H3K27ac HiChIP successfully predicted enhancer elements regulating *AR* and *NKX3-1* gene expressions in LNCaP cells [27]. Therefore, it would be very impactful to decipher similar gene regulation mechanisms in gastric cancer.

The RAS/mitogen-activated protein kinase (MAPK) pathway is one of the most commonly mutated oncogenic pathways in cancer. The MAPK pathway is hyperactivated in several tumors and plays a pivotal role in human cancer development [28,29]. Since MAPK pathway molecules are crucial regulators of drug sensitivity and resistance in cancers, it acts as an attractive target for therapeutic intervention [30]. Mitogen-activated protein kinase 8 (*MAP3K8*) is a vital component of the MAPK pathway and is associated with the development and progression of various cancers. *MAP3K8* could promote myeloma progression by TNFα-mediated ERK activation [31]. ZNF507 mediated transcriptional regulation of *MAP3K8* accelerated the progression of prostate cancer to a highly metastatic and aggressive state [32]. *MAP3K8* also acts as a prognostic biomarker for various cancers including, renal clear cell carcinoma, glioma, and high-grade serous ovarian carcinomas [33,34,35]. Moreover, *MAP3K8* expression is also associated with resistance to chemotherapeutic agents in thyroid cancer and melanoma [36,37]. Because of these wide-ranging effects, *MAP3K8* presents itself as an attractive candidate for further studies in EBVaGC.

In the current study, we have performed ChIP-seq and HiChIP-seq experiments to screen and identify the super-enhancer with chromosomal interaction in EBVaGC and found *MAP3K8* as a potential super-enhancer-regulated gene. Then, we explored the epigenetic regulation and the function of *MAP3K8* in EBVaGC.

## 2. Results

### 2.1. Profiling Super-Enhancer Landscapes in EBVaGC and Identifying MAP3K8 as an SE-Regulated Gene in EBVaGC

Histone-specific landscapes of active chromatin in EBVaGC were identified by ChIP-seq using the H3K27ac antibody. Super-enhancers were then identified by ranking the H3K27ac signals from sliding windows containing enhancers using ROSE. The results indicated that a large number of genes were associated with super-enhancers in EBVaGC (Figure 1A). We integrated H3K27ac ChIP-seq, H3K27ac HiChIP-seq, and RNA-seq data to map common SE-associated genes in EBVaGC. The Venn diagram generated as a result of this integration highlighted the presence of 29 common genes with active super-enhancers (Figure 1B). qRT-PCR was then performed to determine the expression of these genes in gastric epithelial cells and EBVaGC cells. The results revealed that the mRNA levels of *MAP3K8*, *SPINT2*, *FAT1*, *LBR*, *CLDN4*, *PTPN12,* etc., were higher in EBVaGC cells as compared to the gastric epithelial cells (Figure 1C). Further, an extensive literature review verified *MAP3K8* to be a proto-oncogene [33,38,39,40,41]. Moreover, as *MAP3K8* is an integral part of the MAPK cascade, which plays a pivotal role in various cancers’ progression and drug resistance [30], it was selected for further studies.

The H3K27ac profile of *MAP3K8* in SNU719 and YCCEL1 revealed the presence of super-enhancers surrounding the *MAP3K8* gene body (Figure 1D, lower panel). The H3K27ac expression profile obtained in our ChIP-seq experiment was validated by comparing it with the H3K27ac profiles of EBVaGC cells already present in GSE135175 (GES-1 rep1/rep2, YCC10 rep1/rep2, and SNU719 rep1) (Figure 1D, lower panel) [42]. Further, HiChIP-seq data analysis indicated the presence of various long and short-range chromatin interactions of these super-enhancer regions either with the *MAP3K8* gene body or with other active chromatin regions (Figure 1D, upper panel). These results indicate that super-enhances could regulate *MAP3K8* expression in EBVaGC by direct or indirect interaction with the *MAP3K8* gene body.

### 2.2. Super-Enhancers Regulate MAP3K8 Expression in EBVaGC

CRISPR interference (CRISPRi) technology was used to repress four super-enhancer regions associated with *MAP3K8* at the following positions (a) 30,706,373–30,709,258 (Region 1); (b) 30,711,625–30,713,797 (Region 2); (c) 30,716,284–30,717,068 (Region 3); and (d) 30,816,494–30,819,552 (Region 4), respectively, on human chromosome 10, and stable EBVaGC cell lines were generated. The qRT-PCR of *MAP3K8* expression revealed that inhibiting various SE regions repressed *MAP3K8* expression to various extents, ranging between 27% and 55% in SNU719 cells (Figure 2A). CRISPRi mediated repression of the SE at Region 2 produced almost 55% inhibition of *MAP3K8*, whereas repression of the SE at Region 4 resulted in around 27% reduction of *MAP3K8* expression in SNU719 cells (Figure 2A). Similarly, interfering with the activities of Regions 1 and 4 attenuated *MAP3K8* expression by almost 30% in YCCEL1 cells, whereas repression of Regions 2 and 3 cut down *MAP3K8* expressions by nearly 20% with respect to the control cells (Figure 2B).

The stable cell lines were then subjected to MTT assay to determine the effects of SE repression on EBVaGC cell proliferation. The MTT results indicated that the rates of proliferation of *MAP3K8* SE repressed SNU719 cells (Figure 2C) and YCCEL1 cells (Figure 2D) were significantly lower than the control cells. A significant difference in proliferation between the control cells and *MAP3K8* CRISPRi cells can be clearly observed after the 72 h time point (Figure 2C,D). It is very important to mention here that SE-mediated *MAP3K8* regulation might be one of several epigenetic factors controlling *MAP3K8* expression in EBVaGC. The other common ones being DNA methylation and non-coding RNA (ncRNA)-associated gene regulation [43], which calls for further elaborate studies. Moreover, we could also clearly see the varied extent and number of interactions between the *MAP3K8* gene body with other chromosomal regions in SNU719 and YCCEL1 cells in our HiChIP-seq interaction map (Figure 1A, upper panel), which could result in a variability in results between SNU719 and YCCEL1 cells.

Further, we applied the bromodomain and extra-terminal motif (BET) inhibitor (OTX015) to treat EBVaGC cells and examined the effects of OTX015 on *MAP3K8* expression and cellular proliferation by performing qRT-PCR, WB, and proliferation assays. BET domain proteins are readers of acetylated histone residues. BET inhibitors reversibly bind and mask the BET domains, which houses the super-enhancers and regulates the transcription mechanism. The qRT-PCR and WB of OTX015 treated SNU719 cells indicated that BET inhibition could significantly impede *MAP3K8* expression at both the RNA and protein levels in SNU719 (Figure 2E,F). Moreover, MTT assay using OTX015 treated SNU719 cells also revealed that BET motif inhibition could impede the proliferation of SNU719 cells (Figure 2G). In a very similar fashion, OTX015 treatment not only attenuated *MAP3K8* expression in YCCEL1 cells (Figure 2H,I) but also hindered its proliferation (Figure 2J). Taken together, the results indicate that super-enhancers play a vital role in regulating *MAP3K8* expression in EBVaGC.

### 2.3. MAP3K8 Fosters GC Progression by Promoting Cell Proliferation and Migration

Functional analysis of *MAP3K8* was carried out in EBVaGC cells by knocking down *MAP3K8* using short interfering RNA molecules (siRNA). The knockdown efficiency was determined using qRT-PCR and WB. Results indicated that siRNAs could successfully down-regulate *MAP3K8* expression in EBVaGC cells. *MAP3K8* siRNA#1 and *MAP3K8* siRNA#2 were capable of repressing *MAP3K8* expression by almost 40% and 75% in SNU719 cells (Figure 3A). In addition, siRNA treatment attenuated SNU719 cell proliferation (Figure 3B). Similarly, *MAP3K8* siRNA#1 and *MAP3K8* siRNA#2 treatment not only suppressed *MAP3K8* expression by ~68% and ~78% in YCCEL1 cells (Figure 3C), but it also restricted YCCEL1 cell growth (Figure 3D). Further, we also observed that impeding *MAP3K8* could restrict the colony forming ability of SNU719 and YCCEL1 cells. SNU719 cells treated with *MAP3K8* siRNA#1 produced 38% fewer colonies with respect to the control cells treated with scrambled siRNA, whereas, SNU719 cells treated with *MAP3K8* siRNA#2 yielded 67% fewer colonies than the control cells (Figure 3E,F). Similarly, YCCEL1 cells treated with *MAP3K8* siRNA#1 and *MAP3K8* siRNA #2 produced 74% and 79% fewer colonies, respectively, compared to the control cells (Figure 3G,H). Moreover, the results of the Boyden chamber-assisted cell migration assays revealed that *MAP3K8* inhibition could alleviate the migratory potential of EBVaGC cells. The number of SNU719 cells traversing the Boyden chambers reduced by 36% and 69%, respectively, upon treatment with *MAP3K8* siRNA#1 and *MAP3K8* siRNA#2 (Figure 3I,J). Likewise, *MAP3K8* siRNA#1 and *MAP3K8* siRNA#2 treatment also curtailed the migration of YCCEL1 cells by 53% and 93%, respectively (Figure 3K,L).

Taken together, the results highlight the role of *MAP3K8* in EBVaGC progression by the virtue of its ability to promote cell proliferation, migration, etc., and also suggest that *MAP3K8* could act as a potential target for the therapeutic intervention of EBVaGC.

### 2.4. MAP3K8 Inhibits EBVaGC Progression by Inactivating the Notch Signaling Pathway and Epithelial to Mesenchymal Transition (EMT)

RNA-seq data of *MAP3K8* siRNA-treated EBVaGC cells were used for pathway enrichment analysis. Gene set enrichment analysis (GSEA) results indicated that the notch signaling pathway was substantially curtailed upon *MAP3K8* siRNA treatment (Figure 4A). A qRT-PCR analysis of notch pathway-associated molecules, such as *Notch1*, *Notch2*, and C-terminal binding protein 2 (*CTBP2*), etc., revealed that knocking down *MAP3K8* attenuated the expression of *Notch1*, *Notch2*, and *CTBP2* in both SNU719 (Figure 4B, upper panel) and YCCEL1 cells (Figure 4B, lower panel).

Since we have already determined that *MAP3K8* siRNA treatment could restrict the movement of EBVaGC cells traversing the Boyden chamber, we performed a qRT-PCR to detect the expression levels of EMT-associated genes, such as transforming growth factor beta receptor 1 (*TGFβR1*), alpha smooth muscle actin isotype 2 (*ACTA2*), snail family transcriptional repressor 1 (*SNAI1*), and snail family transcriptional repressor 2 (*SNAI2*) [44], in the control and *MAP3K8* siRNA treated EBVaGC cells. Our results indicated that knocking down *MAP3K8* could significantly abrogate the expression of *TGFβR1*, *ACTA2*, *SNAI1*, and *SNAI2* in EBVaGC cells (Figure 4C). These results indicate that *MAP3K8* could regulate EBVaGC progression by deactivating the notch pathway and EMT.

### 2.5. MAP3K8 Plays a Vital Role in GC Progression

Lastly, we checked the expression levels of *MAP3K8* in stomach adenocarcinoma (STAD) in publicly available databases. An analysis of *MAP3K8* expression data from the TCGA and GSE51575 databases revealed that *MAP3K8* was expressed more in cancer tissues as compared to the adjacent control tissues (Figure 5A) and the expression of *MAP3K8* was inversely proportional to the disease-free progression of GC (Figure 5B). The results also revealed that the expression of *MAP3K8* was augmented with the disease stages. Patients with stage 3 and stage 4 STAD had a higher expression of *MAP3K8* in cancer tissues as compared to the patients at stages 2, 1, and the patients in the control group (Figure 5C).

To further substantiate the mechanism underlying *MAP3K8* regulated GC progression, we performed a correlation analysis of *MAP3K8* with the three notch pathway associated molecules and four EMT associated molecules verified by qRT-PCR previously. Spearman’s correlation data revealed that *MAP3K8* expression was positively correlated with the expressions of *Notch1*, *Notch2*, *CTBP2*, *ACTA2*, *TGFβR1*, *SNAI1*, and *SNAI2* with correlation coefficients (R) of 0.25, 0.26, 0.12, 0.13, 0.22, 0.26, and 0.31, respectively (Figure 5D). We also observed that the expressions of *TGFβR1*, *SNAI1*, and *CTBP2* were higher in GC (Appendix A, upper panel) and inversely correlated with the disease-free progression (Appendix A, lower panel). These data further confirm that *MAP3K8* plays an important role in gastric cancer progression, probably through the activation of the notch signaling pathway and EMT.

## 3. Discussion

The mitogen-activated protein kinase (MAPK) pathway is one of the most well-defined and widely studied pathways in cancer biology. Hyperactivation of MAPK signaling is associated with more than 40% of human cancer cases [45]. Mitogen-activated protein kinase 8 (*MAP3K8)* is an indispensable member of the MAPK family and its overexpression has been widely implicated in the development and progression of numerous cancers, including prostate cancer, colorectal cancer, lung cancer, nasopharyngeal carcinoma, breast cancer, etc. [46,47,48,49,50]. Moreover, *MAP3K8* activation is also known to be a critical driving factor for drug resistance in various cancers, including melanoma, chronic myeloid leukemia, and thyroid cancer stem cells [36,37,51]. *MAP3K8* proto-oncogene activation could trigger the MAPK cascade and regulate the MEK/ERK/JNK, TLR, and IKK/NFKB pathway responses, which not only play decisive roles at different stages of tumor development but also affect immune surveillance and responses to anti-cancer therapy [38,52,53]. Therefore, because of the multiple pathways regulated by *MAP3K8* in cancer progression, *MAP3K8* and the MAPK cascade have remained attractive targets for new cancer therapy targets.

Analysis of previous RNA-seq data (GSE51575) indicated that *MAP3K8* was differentially expressed in GC. TCGA data analysis also revealed that *MAP3K8* expression was higher in GC tissues as compared to the adjoining normal tissues and it inversely correlated with the disease/progression-free survival of GC patients. The results are in line with previous reports on other cancers. The TCGA database analysis further revealed that *MAP3K8* expression was directly associated with stomach adenocarcinoma stage, with patients in stages 3, 2, and 4 of stomach cancer expressing more *MAP3K8* compared to the patients in stage 1 of the disease and normal patients. Additionally, a functional analysis of *MAP3K8* in EBVaGC cells revealed that *MAP3K8* was essential for the survival, proliferation, colony formation, and migration of SNU719 and YCCEL1 cells. Taken together, our results signify the role of *MAP3K8* in EBVaGC progression and also that *MAP3K8* can act as a potential target for therapeutic intervention.

Over the past decade, numerous reports have highlighted the increasing role of epigenetics in oncogenesis. Epigenetic processes, such as DNA methylation, histone modification, and various RNA-mediated processes have been reported to influence gene expression predominantly at the transcription levels; however, other steps in the process may also be regulated epigenetically [54]. Epigenetically regulated gene expression plays a crucial role in colorectal cancer progression [9,55]. Similarly, numerous tumor-associated genes, including *COL1A2, PD-L1, HOXB* cluster genes, and *CLCA2,* are epigenetically regulated in breast cancer cells [56,57,58,59]. Furthermore, Li et al. demonstrated that hypermethylation of the CREB binding motif of the *DAZAP2* gene in multiple myeloma cells promoted tumor progression by p38/MAPK pathway activation [60]. Raedt et al. showed that *SUZ12* inactivation triggered an epigenetic switch, sensitizing cancer cells to bromodomain inhibitors [61].

The BET domain has emerged as an attractive target controlling the epigenetic regulation of tumor-associated genes. Previous studies have indicated that inhibiting the BET domain attenuated cancer progression by mitigating *PD-L1* expression and HGF-MET signaling [62,63]. Additionally, the noncoding RNAs could directly target the epigenetic machinery, thereby altering the expression of tumor suppressor genes or oncogenes [64,65]. Gastric cancer progression is frequently associated with the epigenetic regulation of numerous cancer-related genes and long noncoding RNAs. Hypermethylation of the *DCBLD2* gene promoter stimulated cancer cell proliferation and the invasion of SNU cell lineages [66]. Similarly, aberrant methylation of the *PKD1* gene promoter influenced migration and metastasis in gastric cancer [67]. Zhang et al. reported that methylation of lncRNA KCNK15-AS1 promoted gastric cancer progression by modulating the KCNK15-AS1-DNMT1-MAPK and KCNK15-AS1-HDAC1-AKT axes, and treatment with 5-azacytidine and chidamide reversed this effect [68]. Similarly, many more lncRNA have also been implicated in the development and progression of GC [69].

In the current study, we have observed differential H3K27ac levels in four regions surrounding the *MAP3K8* gene on human chromosome 10. The contrasting H3K27ac expression between gastric epithelia and EBV-positive GC cells indicates the presence of active super-enhancers of *MAP3K8* in EBVaGC. To substantiate the effect of these active super-enhancers on MAP3K8 expression, EBVaGC cells were transfected with CRISPRi plasmids targeting these SE regions and it was found that suppressing the SE regions attenuated *MAP3K8* expression. The MTT assay further revealed that suppressing *MAP3K8* stifled EBVaGC cell proliferation. Additionally, we also validated the above results by treating SNU719 and YCCEL1 cells with birabresib (OTX015), which specifically binds to and inhibits the BET domain containing the SE regions. OTX015 treatment not only attenuated *MAP3K8* expression in EBVaGC cells at both the RNA and protein levels but also impeded their proliferation. Taken together, these results indicate that the super-enhancers play a considerable role in regulating *MAP3K8* expression in EBVaGC. Further, the functional analysis of *MAP3K8* in EBVaGC progression revealed that *MAP3K8* knockdown attenuated the cell proliferation, colony formation, and migration of EBV-positive GC cells.

The notch signaling pathway not only plays a pivotal role in the self-renewal of stem cells but is also involved in the cell-fate determination of progenitors [70]. In addition, notch signaling has been reported to be highly expressed and activated in gastric cancer [71]. A meta-analysis of 15 studies containing 1547 gastric cancer cases and 450 controls from the MEDLINE, EMBASE, and Chinese National Knowledge Infrastructure (CNKI) databases revealed that the expression of *Notch1* and *Notch2* was significantly higher in tumor tissues of GC compared to normal tissues [72]. Notch activation triggers gastric stem/progenitor cell proliferation and GC tumor formation by activating the Notch1 and Notch2 receptors [73]. A recent study demonstrated that Penicilazaphilone C could induce apoptosis in gastric cancer by blocking the notch receptors, Notch1 and Notch2 [74]. Furthermore, overexpression of the transcriptional co-repressor C-terminal binding protein-2 is closely correlated with advanced tumor stage and poor prognosis in GC patients. *CTBP2* could induce epithelial to mesenchymal transition, and silencing *CTBP2* could impede GC growth in nude mice model [75]. Our results indicate that *MAP3K8* repression could inhibit EBVaGC progression by attenuating the expression of the notch receptors, Notch1 and Notch2, and also by regulating the expression of C-terminal binding protein-2.

Epithelial to mesenchymal transition is considered to be a major factor underlying cancer progression [76,77]. Ishimoto et al. reported that the invasiveness of gastric cancer cells is associated with the activation of *TGFBR1* signaling [78]. Additionally, well-known transcription factors, such as SNAI1 and SNAI2, play a crucial role in promoting GC progression. A recent report suggested that *USP37*-mediated deubiquitination of *SNAI1* could promote the proliferation and migration of gastric cancer cells [79]. Similarly, GC metastasis and tumor growth were also associated with the *CXCR4-SNAI2* signaling pathway [80].

It has also been reported that mesenchymal cells formed during carcinogenesis express substantial amounts of α-SMA/*ACTA2* [81]. Numerous studies have highlighted the role of α-SMA in cancer progression. *ACTA2* has been reported to promote invasion and metastasis in lung adenocarcinoma by regulating *c-MET* and *FAK* expression [82]. Additionally, α-SMA expression is also associated with poor prognosis and survival in various cancers, including gastric carcinoma, nasopharyngeal carcinoma, and oral tongue squamous cell carcinoma (OTSCC) [83,84,85]. Our qRT-PCR data reveal that suppressing *MAP3K8* curtailed *TGFBR1*, *SNAI1*, *SNAI2*, and *ACTA2* expression in both SNU719 and YCCEL1 cells. Moreover, correlation studies of *MAP3K8* with *TGFBR1*, *SNAI1*, *SNAI2*, and *ACTA2* showed that the expression of these genes was positively correlated with *MAP3K8* expression in GC. Based on the above results, we could conclude that repressing *MAP3K8* could inhibit EBVaGC progression by restricting the notch signaling pathway, and also epithelial to mesenchymal transition.

Although our current observations indicate the involvement of the notch pathway and EMT in *MAP3K8* modulated EBVaGC progression, a direct/indirect link between *MAP3K8* expression and the notch–EMT axis needs to be established. Therefore, further validation of our current findings are needed via detailed study of *MAP3K8* expression on the notch–EMT axis by determining if notch inactivation downstream of *MAP3K8* down-regulation is necessary or sufficient to confer growth or migration inhibition, and whether enforcing notch signaling after *MAP3K8* knockdown rescues the growth defect. In addition, it is also necessary to determine if *MAP3K8* depletion alters the functional markers of EMT, such as cell polarity, morphology, or the nuclear localization of Snail or Twist.

## 4. Materials and Methods

### 4.1. Cell Culture

Epstein–Barr virus-associated human gastric carcinoma cell lines (SNU719 and YCCEL1), gastric epithelial cells (GES-1), and HEK293T cells were obtained from the State Key Laboratory of Oncology, Sun Yat-sen University Cancer Center, Guangzhou, China. EBVaGC cells and GES-1 were grown in Roswell Park Memorial Institute (RPMI) 1640 medium whereas the growth medium for the HEK293T cells was Dulbecco’s modified Eagle’s medium (DMEM). Cell culture mediums were supplemented with 10% FBS, and 1% penicillin and streptomycin solution. The cells were maintained in a humidified atmosphere of 5% CO_2_ at 37 °C. All cell culture reagents were purchased from Life Technologies, Inc. (MD, USA), unless mentioned specifically.

### 4.2. H3K27ac ChIP-seq

For the GES-1, SNU719, and YCCEL1 cell lines, 15 × 10^6^ cells were cross-linked with 1% formaldehyde and lysed. The genomic DNA was sonicated using an ultrasonicator (Covaris E220) (Covaris, MA, USA) to produce around 500 bp fragments in the lysis buffer. After clearing the debris, the supernatants were diluted, and the lysates were incubated with anti-H3K27ac antibody (Abcam, Cambridge, UK, catalogue no.: ab4729) overnight at 4 °C. Protein A agarose beads were then used to capture the protein–DNA complexes. After an extensive wash, the protein–DNA complexes were eluted and reverse cross-linked. The DNA was then purified using the Qiagen PCR purification kit (Qiagen, Hilden, Germany, catalogue no.: 28106). Further, a NEBNext ChIP-Seq library preparation kit for Illumina (NEB, Ipswich, MA, USA, catalogue no.: E7645) was used to prepare the DNA libraries using 10 ng DNA as the starting material. The sequencing libraries were analyzed by an Equalbit dsDNA HS Assay Kit (Vazyme, Nanjing, China, catalogue no.: EQ111) and quantified via qPCR for quality control. The libraries were sequenced using an Illumina HiSeq X10. Inputs from each sample were sequenced together using different barcodes.

### 4.3. H3K27ac HiChIP-seq

For the SNU719 and YCCEL1 cell lines, 15 × 10^6^ cells were cross-linked with 1% formaldehyde and lysed with Hi-C lysis buffer (10 mM Tris-HCl pH 7.5, 10 mM NaCl, 0.2% NP-40, 1X Roche protease inhibitor cocktail). The cross-linked DNA were digested by 375 U MboI (NEB, catalogue no.: R0147M) at 37 °C for 2 h. DNA ends were filled in with a biotin-dATP (Thermo Fisher Scientific, MA, USA, catalogue no.: 19524016), dCTP, dGTP, and dTTP mix, and a DNA polymerase I (Large) Klenow fragment (NEB, catalogue no.: M0210) with constant shaking at 37 °C for 1 h. T4 DNA ligase (NEB, catalogue no.: M0202L) was used for proximity DNA ligation at 4 °C overnight. After ligation, the DNA was sonicated, diluted 10 times with ChIP dilution buffer (0.01% SDS, 1.1% Triton X-100, 1.2 mM EDTA, 16.7 mM Tris-HCl pH 7.5, 167 mM NaCl) and precleared with Protein A Dynabeads (Invitrogen, CA, USA, catalogue no.: 1001D) at 4 °C for 1 h. The DNA–protein complexes were captured with 8 μL H3K27ac antibody at 4 °C overnight. The DNA–protein complexes were then captured with Protein A beads with a rotation at 4 °C for 2 h. The beads were washed, and the DNA was eluted twice with 150 μL of freshly prepared DNA elution buffer (50 mM sodium bicarbonate pH 8.0, 1% SDS). The eluted DNA was reverse cross-linked and purified using a PCR purification Kit (Qiagen, catalogue no.: 28106). Biotin dATP labeled DNA was captured with 5 μL Streptavidin C-1 beads (Thermo Fisher Scientific, catalogue no.: 65001), and the DNA was then transposed with 2.5 μL of Tn5. The beads were then washed and re-suspended with 23 μL ddH_2_O, 25 μL 2X Phusion HF (NEB, catalogue no.: M0531S), and 1 μL each of Nextera forward primer (Ad1_noMX) and Nextera reverse primer (Ad2.X) at 12.5 μM. A PCR was run at 72 °C for 5 min, 98 °C for 1 min, 98 °C for 15 s, and 63 °C for 30 s, for a total of 8 cycles, and a final extension at 72 °C for 1 min. After PCR amplification, a two-sided size selection with AMPure XP beads (Beckman Coulter, CA, USA, catalogue no.: A63381) was performed for DNA size selection and purification. The samples were then sequenced on the Illumina Nextseq platform (2 × 75 bp).

### 4.4. ChIP-seq and HiChIP-seq Data Analysis

The ChIP-seq and HiChIP-seq sequencing reads were subjected to FastQC (https://www.bioinformatics.babraham.ac.uk/projects/fastqc (accessed on 11 January 2023)) to ensure that the sequencing experiments had no considerable flaws, such as a heavy GC bias and PCR artifacts.

The ChIP-seq reads were aligned to the human (hg19) genome using *Bowtie2* v2.2.373 under default settings. The read mappability rate ranged from 94% to 98% across the ChIP-seq samples. ChIP-seq peaks were called using *MACS* v2.1.074 with default settings on each sample. Peaks located in blacklist regions were excluded from the analysis. The genome-wide ChIP-seq coverage was normalized with size factors, which were determined using *DiffBind* combined with *DESeq2*. SEs were called with *ROSE* v1.0.0 under default settings using H3K27ac peaks from the ChIP-seq data. The normal used for EBVaGC specific super-enhancers selection was normal gastric cells GES-1. To identify the EBVaGC specific super-enhancer, all EBV enhancers were ranked according to their total background-subtracted H3K27ac ChIP-seq signal. H3K27ac signals within 12.5 kb were stitched together. EBVaGC enhancers were sorted and plotted based on the H3K27ac signals in ascending order. The X-axis shows the H3K27ac ChIP-seq signals rank order; the Y-axis shows normalized H3K27ac signals. A line was drawn from the first enhancer with the lowest signal to the last enhancer with the highest signal to determine a diagonal slope. A point on the ranked plot with a tangent line identical to the diagonal slope was identified. This X-axis point was set as the cut-off to distinguish EBVaGC super-enhancers from EBVaGC typical-enhancers. The EBV-GC enhancers with H3K27ac signals higher than this point were assigned as EBVaGC super-enhancers. Super-enhancers of normal gastric cells GES-1 were identified in the same way. Then, the EBVaGC specific super-enhancers were determined with the exclusion of the super-enhancers of normal gastric cells GES-1. Finally, we checked the 3D enhancer-promoter loops within the EBVaGC specific super-enhancers region and found that *MAP3K8* was significantly regulated by EBVaGC specific super-enhancers.

HiChIP-seq paired-end reads (17–27 million reads for each sample) were mapped using *HiC-Pro* v2.11.133 (default settings with LIGATION_SITE set as GATCGATC for MboI) and significant loops were identified with *hichipper* v0.7.534 (default settings except parameter—skip-diffloop was set). In detail, HiChIP-seq experiments were performed with MboI, which can introduce a bias to the fragment sizes due to the non-uniform distribution of cut sites. The *hichipper* performs a background correction based on the non-uniform distribution of restriction fragments, to better infer anchors and loops. Quality control was performed by ensuring a high percentage of reads were mapped (>85%), within anchors (>70%), and supporting valid interactions (18–30%). Mapped reads were also visually inspected on the WashU genome browser to ensure significant enrichment when compared to the H3K27ac ChIP-seq reads, general uniformity between replicates, and a good signal-to-noise ratio, which are indicative of a successful ChIP experiment. Replicate samples were then merged, and loops were identified by (1) merging anchors within 1.5 kb of each other, and (2) removing loops that were <5 kb in length. Loops scores were further normalized by the total number of valid interaction read pairs in each cell line. A minimum of three normalized read pairs were used to filter strong long-range interactions. Long-range interactions were then annotated using *diffloop* v1.10.081 (default parameters with enhancer and promoter regions defined as below) to decide the enhancer–enhancer loops, enhancer–promoter loops, and promoter–promoter loops. Here, promoters were defined as ± 2 kb regions surrounding gene transcription start sites. Enhancers were defined as identified H3K27ac binding regions except those located at promoter regions.

### 4.5. Integrative Analysis

Differentially expressed genes (DEGs) in gastric cancer tissue (GSE51575) (https://www.ncbi.nlm.nih.gov/geo/query/acc.cgi?acc=GSE51575 (accessed on 11 January 2023)) and gastric cancer cell lines (GSE147152) (https://www.ncbi.nlm.nih.gov/geo/query/acc.cgi?acc=GSE147152 (accessed on 11 January 2023)) were included in combination with the H3K27ac ChIP-seq and HiChIP-seq data for further screening of key regulatory genes in gastric carcinoma. The differentially expressed genes in GC tissues were identified using adjusted p value < 0.05 and logFC > 0.5 as the threshold and the DEGs in the gastric cancer cell lines (SNU719_RNAseq_rep1/rep2 and YCC10_RNAseq_rep1/rep2) were identified using adjusted p value < 0.005 and logFC > 1.5 as the threshold. Further, for the HiChIP-seq data of EBVaGC, two replicates each of YCCEL1 and SNU719 cell lines were used. Genes flanking the HiChIP anchor loci were identified as target genes from the HiChIP-seq data. Finally, the ChIP-seq data were used to identify the SEs in GC, as mentioned in Section 2.5. Super enhancer-regulated genes (SEGs) in GC were ascertained by subtracting the genes flanking the EBVaGC-specific super-enhancer regions (SNU719 and YCCEL1) from the super-enhancer-flanked genes in the normal gastric epithelium cell line (GES-1). The data were integrated to produce a Venn diagram showing overlaps representing SE-regulated genes in EBVaGC.

### 4.6. Plasmid Construction, Transfection, and Establishment of MAP3K8 SE-CRISPRi Stable Cell Lines

Plasmid lenti_dCas9-KRAB-MeCP2 was procured from Addgene (WA, USA) and the CRISPRi constructs were designed using the CHOPCHOP (http://chopchop.cbu.uib.no/ (accessed on 11 January 2023)) guide design tool (Appendix A). Lenti_dCas9-KRAB-MeCP2 plasmid was digested using BsmBI-v2 (NEB, catalogue no.: R0739), purified on an agarose gel, and the CRISPRi constructs were incorporated into the linearized plasmid using T4 DNA ligase (Thermo Fisher Scientific, catalogue no.: EL0011). HEK293T cells were then transfected with CRISPRi plasmid, lentiviral packaging plasmid (psPAX2, Addgene, catalogue no.: 12260), and VSV-G envelope expressing plasmid (pMD2.G, Addgene, catalogue no.: 12259), simultaneously, using a lipofectamine 3000 transfection reagent (Thermo Fisher Scientific, catalogue no.: L3000-008). Supernatant containing the virus particles were collected after 48 h. The EBVaGC cells were then treated with the virus particles supplemented with 2 µg/mL polybrene (Sigma-Aldrich, MO, USA, catalogue no.: TR-1003-G). The cells were supplemented with fresh medium after 12 h. The cells were then put under puromycin selection (0.7 µg/mL, Thermo Fisher Scientific, catalogue no.: A11138-03) for at least 7 days to produce *MAP3K8* SE-repressed stable EBVaGC cells.

### 4.7. siRNA Transfection

Small Interfering RNAs (siRNA) against *MAP3K8* were purchased from Guangzhou RiboBio Co., Ltd., (Guangzhou, China) and were diluted in ddH_2_O according to the manufacturer’s instructions. Transfection was performed using the Lipofectamine^TM^ RNAiMAX transfection reagent (Thermo Fisher Scientific, catalogue no.: 13778150). Briefly, siRNA was mixed with RNAiMAX and added to 40–50% confluent cell monolayers in Opti-MEM medium (GIBCO, MA, USA). The minimal essential medium was replaced with RPMI 1640 containing 10% FBS after 6 h and the cells were allowed to grow for another 48 h. Transfection efficiency was then determined using qRT-PCR and WB, and the cells were also used to perform functional analyses, such as an MTT, colony formation assay, and migration assay.

### 4.8. Birabresib (OTX015) Treatment

OTX015 (Selleckchem, TX, USA, catalogue no.: S7360) was dissolved in dimethyl sulphoxide (DMSO) and used to treat EBVaGC cells.

### 4.9. Cell Viability Assay

An MTT assay was performed to determine EBVaGC cell viability. In brief, 3000 cells were seeded per well in 96-well plates and the number of viable cells were determined at various time points ranging from 24 h, 48 h, 72 h, 96 h, and 120 h, respectively. An MTT (Sigma-Aldrich, catalogue no.: M2128) was added, and the optical density (OD) of the isopropanol dissolved formazan crystals was measured after 3.5–4 h at 570 nm using the SpectraMax 190 microplate reader (Molecular Devices, CA, USA). The experiments were performed in quadruplet and all experiments were repeated thrice.

### 4.10. Colony Formation Assay

For colony formation assays, 300 cells of the SNU719 lineage and 500 cells of the YCCEL1 lineage were seeded per well in 35 mm cell culture dishes or 6-well plates. The cells were allowed to grow in a controlled humidified environment for 10–12 days, with a media change every 2–3 days. Cells were then washed in PBS, fixed using methanol, and stained with 0.5% crystal violet (Amresco, OH, USA, catalogue no.: 0528) solution. Images of the cell colonies were taken using a Bio-Rad ChemiDoc Imaging system (Bio-Rad, Hercules, CA, USA). Each experiment was carried out in triplicate and repeated at least three times.

### 4.11. Boyden Chamber Assisted Cell Migration Assay

Cell migration assays were performed to determine the effect of *MAP3K8* knockdown on the motility of EBVaGC cells. Briefly, 2 × 10^4^ control and *MAP3K8* knock-down EBVaGC cells were seeded on the top compartment of the transwell chambers in basal RPMI 1640. RPMI 1640 containing 10% FBS acted as a chemo-attractant and was placed at the bottom wells. The cells were allowed to migrate under the influence of the chemo-attractant for another 24 h. Cells were then fixed using chilled methanol for 20 min, washed with PBS, and stained with 0.05% crystal violet in 20% methanol for 30 min. The transwells were washed thrice using PBS and the cells on the upper part of the transwell chambers/non-migrated cells were removed by gently dabbing with cotton swabs. The transwells were allowed to air-dry and images were taken at 100× magnification using a camera fitted inverted microscope. The number of migrated cells was counted manually in three high power fields. A minimum of three independent experiments were performed.

### 4.12. Immunoblotting

EBVaGC cells were lysed in 1X Laemmli SDS-PAGE buffer supplemented with 1X protease inhibitor (TargetMol, Boston, MA, USA, catalogue no.: C0001). Total protein was measured using a Pierce^TM^ BCA protein assay kit (Thermo Fisher Scientific, catalogue no.: 23225) and an equal amount of proteins were separated on an SDS-PAGE gel. The proteins were then blotted onto a PVDF membrane (0.4 μm) (Merck Millipore, Burlington, MA, USA, catalogue no.: IPVH00010). Transfer was carried out at 250 mA for 2 h at 4 °C. The membrane was then blocked using 5% non-fat milk in TBS-T (0.1% Tween 20 in TBS) for 2 h at room temperature. Following three washes with TBST, the membrane was incubated overnight at 4 °C with an MAP3K8 primary antibody (Abcam, catalogue no.: ab137589). The membrane was washed thrice with TBS-T for 10 min each and further incubated with a secondary antibody (β-actin) for another 45 min. Following another round of TBST washes (3×, 10 min each), the chemiluminescence signals were visualized using the BioRad ChemiDoc imaging system. MAP3K8 and β-actin antibodies were used at dilutions of 1:1000 and 1:3000, respectively, whereas, both secondary antibodies were used at 1:3000 dilution. The experiments were repeated thrice.

### 4.13. Quantitative Reverse-Transcriptase Polymerase Chain Reaction Analysis (qRT-PCR)

The total RNA from cells was extracted using TRIzol^TM^ reagent (Thermo Fisher Scientific, catalogue no.: 15596-018). cDNA was synthesized from RNA using the GoScript^TM^ reverse transcription system (Promega, Madison, WI, USA, catalogue no.: A5001). Briefly, 1.5 μg total RNA was taken as a template and cDNA synthesis was carried out as per the manufacturer’s instructions. The cDNA was diluted 10 times and was then used for the qRT-PCR experiment to determine gene expressions in EBVaGC cells. The PCR was performed in a LightCycler 480 RT-PCR system (Roche Molecular Systems Inc., Basel, Switzerland) using a ChamQ universal SYBR qPCR master mix (Vazyme, catalogue no.: Q711-02). All qRT-PCR experiments were performed in triplicate and each experiment was performed at least three times. The sequence of the primers used in this study is given in Appendix A.

### 4.14. RNA Sequencing and Pathway Analysis

RNA isolated from MAP3K8 siRNA#1, MAP3K8 siRNA#2, and NCsiRNA treated EBVaGC cells were subjected to RNA sequencing. RNA sequencing was performed by Berry Genomics (Beijing, China). The raw data of the paired-end sequencing was cleaned by *fastp* and then aligned to the UCSC hg19 reference genome using *STAR* to obtain the original BAM format file. The original gene matrix was derived using *featureCounts* and differential gene analysis was performed using *DESeq2*. Finally, GSEA software was used for further analysis using the NOTCH gene set M7946, accessible at www.gsea-msigdb.org/gsea/msigdb/cards/KEGG_NOTCH_SIGNALING_PATHWAY (accessed on 5 May 2022).

### 4.15. Web-Based Analysis of GC Omics Data

Online analysis portals, such as the Kaplan–Meier plotter (https://kmplot.com/analysis/index.php?p=service&cancer=gastric (accessed on 11 January 2023)), UALCAN (the University of Alabama at Birmingham cancer data analysis portal, http://ualcan.path.uab.edu/cgi-bin/TCGAExResultNew2.pl?genenam=MAP3K8&ctype=STAD (accessed on 11 January 2023)), and GEPIA (gene expression profiling interactive analysis, http://gepia.cancer-pku.cn/detail.php?clicktag=correlation (accessed on 11 January 2023)) were used to perform the meta-analysis-based discovery and validation of markers in GC. For the survival analysis, patients were split by median expression of target genes and the Kaplan–Meier survival curve was plotted. The correlation between genes of interest in GC was determined using Spearman’s correlation coefficient.

### 4.16. Statistical Analysis

Statistical analysis was performed using GraphPad Prism 8.0 (GraphPad, CA, USA). Data are represented as the mean ± SD. An unpaired parametric t-test with Welch correction and a two-way ANOVA with Geisser–Greenhouse correction were used to determine the statistical significance between the groups and values of *p* ≤ 0.05 were considered significant. For all experiments, ns = non-significant, *** *p* ≤ 0.001, ** *p* ≤ 0.01, and * *p* ≤ 0.05.

## 5. Conclusions

Our results indicate that super-enhancers regulate *MAP3K8* expression in EBV-associated gastric carcinoma. The epigenetic regulation of *MAP3K8* could control gastric carcinoma progression by modulating the notch pathway and EMT. Taken together, *MAP3K8* inhibition might present an attractive treatment strategy for EBVaGC.

## Figures and Tables

**Figure 1 ijms-24-01964-f001:**
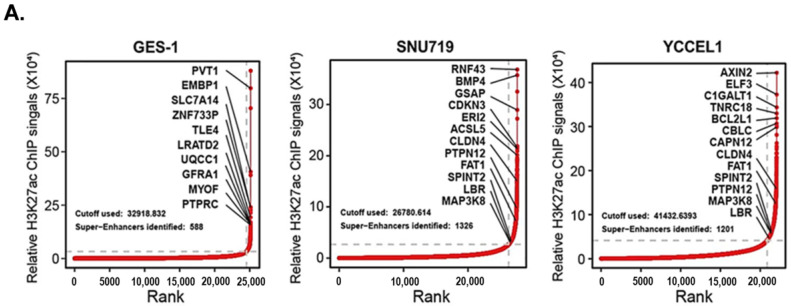
Profiling super-enhancer (SE) landscapes and identifying *MAP3K8* as an SE-regulated gene in Epstein–Barr virus-associated gastric carcinoma (EBVaGC). (**A**) Hockey stick plots in gastric cancer (GC) cells showing input-normalized, rank-ordered H3K27ac signals, highlighting several SE-associated genes. Enhancers are ranked by decreasing H3K27ac ChIP-seq signals. The *X*-axis shows the H3K27ac ChIP-seq signals rank order whereas the *Y*-axis shows normalized H3K27ac signals. The gray dashed line demarcates the boundary between typical enhancers and SEs. (**B**) Venn diagram showing the integrative analysis of RNA-seq, ChIP-seq, and HiChIP-seq data to nominate novel SE-associated genes in GC. Twenty-nine genes with active SE regions overlapped in this analysis. GC-CELL and GC-TISSUE represent mRNA expression profiles of EBV +ve GC cell lines and EBV +ve GC obtained from GSE147152 and GSE51575, respectively. (**C**) Relative mRNA levels of SE-associated genes in GES-1, SNU719, and YCCEL1 cells. Gene expression levels were normalized to the housekeeping gene *GAPDH*. qRT-PCR was performed in triplicate and the mRNA expression profile map was generated using *pheatmap* R package. (**D**) H3K27ac ChIP-seq (lower panel) and HiChIP-seq interaction profiles (upper panel) of *MAP3K8* in SNU719 and YCCEL1 cells. Red arrows point to the regions with differential H3K27ac expression in EBVaGC (lower panel). The *MAP3K8* gene body is highlighted in yellow and the purple loops represent chromosomal interactions (upper panel).

**Figure 2 ijms-24-01964-f002:**
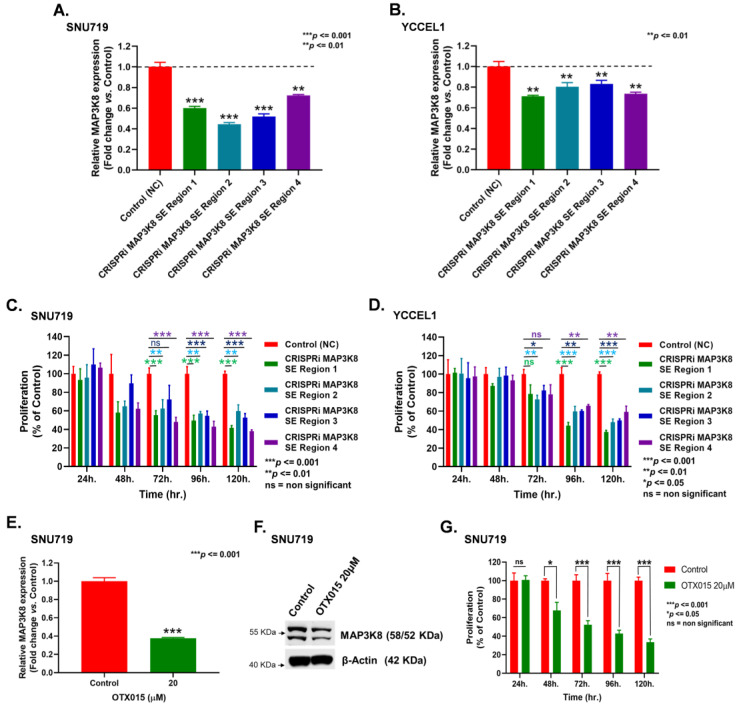
Super-enhancers regulate *MAP3K8* expression in EBVaGC. (**A,B**) Four CRISPRi plasmids corresponding to four *MAP3K8* SE regions located between (a) 30,706,373–30,709,258 kb (Region 1); (b) 30,711,625–30,713,797 kb (Region 2); (c) 30,716,284–30,717,068 kb (Region 3); and (d) 30,816,494–30,819,552 kb (Region 4) of chromosome 10 were formulated. EBVaGC cells were transfected with these plasmids to create *MAP3K8* SE repressed EBVaGC stable cell lines. RNA was isolated 48 h after selection. The relative mRNA levels of *MAP3K8* in *MAP3K8* SE repressed SNU719 (**A**) and YCCEL1 (**B**) cells using qRT-PCR. *MAP3K8* expression levels were normalized to the housekeeping gene *GAPDH*. (**C**,**D**) Proliferation of *MAP3K8* SE repressed SNU719 (**C**) and YCCEL1 (**D**) cells were determined by MTT. An MTT was performed at 24 h, 48 h, 72 h, 96 h, and 120 h, respectively. (**E**,**F**) mRNA (**E**) and protein (**F**) levels of MAP3K8 in OTX015 treated SNU719 cells. SNU719 cells were treated with 20 µM OTX015 and RNA and the total proteins were isolated at the same time (48 h post-treatment) for PCR and WB. (**G**) Proliferation of OTX015 treated SNU719 cells. SNU719 cells were treated with 20 µM OTX015 and an MTT was performed at 24 h, 48 h, 72 h, 96 h, and 120 h, respectively. (**H**,**I**) mRNA (**H**) and protein (**I**) levels of MAP3K8 in OTX015 treated YCCEL1 cells. YCCEL1 cells were treated with 20 µM OTX015 and RNA and total proteins were isolated simultaneously (48 h post-treatment) for PCR and WB. (**J**) Proliferation of OTX015 treated YCCEL1 cells. YCCEL1 cells were treated with 20 µM OTX015 and an MTT was performed at 24 h, 48 h, 72 h, 96 h, and 120 h, respectively. Three and four biological repeats were performed for the qRT-PCR and MTT, respectively. All experiments were repeated a minimum of three times and the data are represented as the mean ± SD (ns = non-significant, *** *p* ≤ 0.001, ** *p* ≤ 0.01, and * *p* ≤ 0.05).

**Figure 3 ijms-24-01964-f003:**
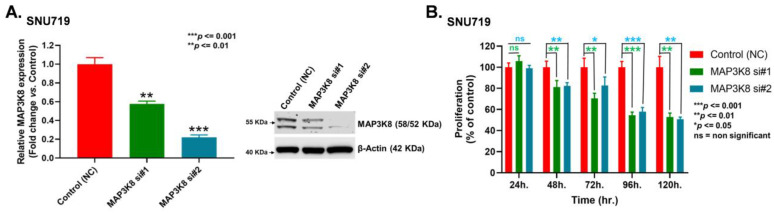
Functional analysis of *MAP3K8* in EBVaGC. (**A**) The knock-down efficiency of *MAP3K8* in SNU719 cells was determined by qRT-PCR (left) and WB (right) at 48 h after siRNA transfection. *GAPDH* and *β-actin* were used as controls for PCR and WB, respectively. (**B**) Proliferation of *MAP3K8* siRNA treated SNU719 cells. MTT readings were taken at 24 h, 48 h, 72 h, 96 h, and 120 h, respectively. (**C**) The knock-down efficiency of *MAP3K8* in YCCEL1 cells was determined by qRT-PCR (left) and WB (right) at 48 h after siRNA transfection. *GAPDH* and *β-actin* were used as controls for the PCR and WB, respectively. (**D**) Proliferation of *MAP3K8* siRNA treated YCCEL1 cells was determined by MTT performed at 24 h, 48 h, 72 h, 96 h, and 120 h, post seeding, respectively. (**E**–**H**) Colony formation assay demonstrated the long term proliferating capability of *MAP3K8* knock-down SNU719 cells (**E**,**F**) and YCCEL1 cells (**G**,**H**). Colonies were stained after a minimum of 10 days. The number of colonies in SNU719 cells (**F**) and YCCEL1 cells (**H**) were quantified. (**I**–**L**) Boyden chamber migration assays were performed to determine the migration ability of *MAP3K8* knockdown SNU719 cells (**I**,**J**) and YCCEL1 cells (**K**,**L**). The number of migrated cells in SNU719 (**J**) and YCCEL1 (**L**) was detected after 24 h and quantified. Images of the emigration assays were taken at 100× magnification and the scale bar was 100 µm. Migration assay graphs were constructed after counting the average number of migrated cells in three high power microscopic fields. MTT was performed in quadruplet, and three biological repeats were performed for all qRT-PCR assays. All experiments were performed thrice and the data are represented as the mean ± SD (ns = non-significant, *** *p* ≤ 0.001, ** *p* ≤ 0.01, and * *p* ≤ 0.05).

**Figure 4 ijms-24-01964-f004:**
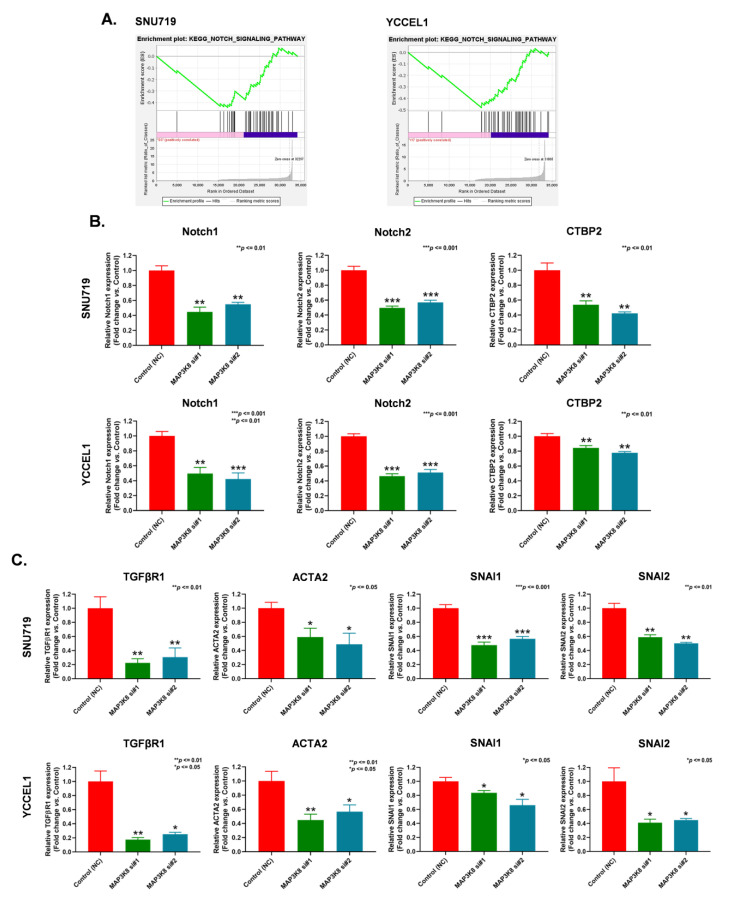
*MAP3K8* impedes EBVaGC progression by down-regulating the notch pathway and EMT. (**A**) GSEA analysis of notch signaling pathway genes in *MAP3K8* suppressed SNU719 (left) and YCCEL1 (right) cells. (**B**) Validation of selective notch signaling pathway genes, such as *Notch1*, *Notch2*, and *CTBP2* by qRT-PCR in *MAP3K8* silenced SNU719 cells (top), and YCCEL1 cells (bottom). (**C**) mRNA expression profiles of *TGFβRI*, *ACTA2*, *SNAI1*, and *SNAI2* in *MAP3K8* knock-down SNU719 cells (top) and YCCEL1 cells (bottom). For all qRT-PCR experiments, 1.5 µg RNA was reverse transcribed to produce cDNA, and diluted 1:10 for the PCR. Gene expressions were normalized to the housekeeping gene *GAPDH*. The qRT-PCR experiments were performed in triplicate, a minimum of three times and the data are represented as the mean ± SD (*** *p* ≤ 0.001, ** *p* ≤ 0.01, and * *p* ≤ 0.05).

**Figure 5 ijms-24-01964-f005:**
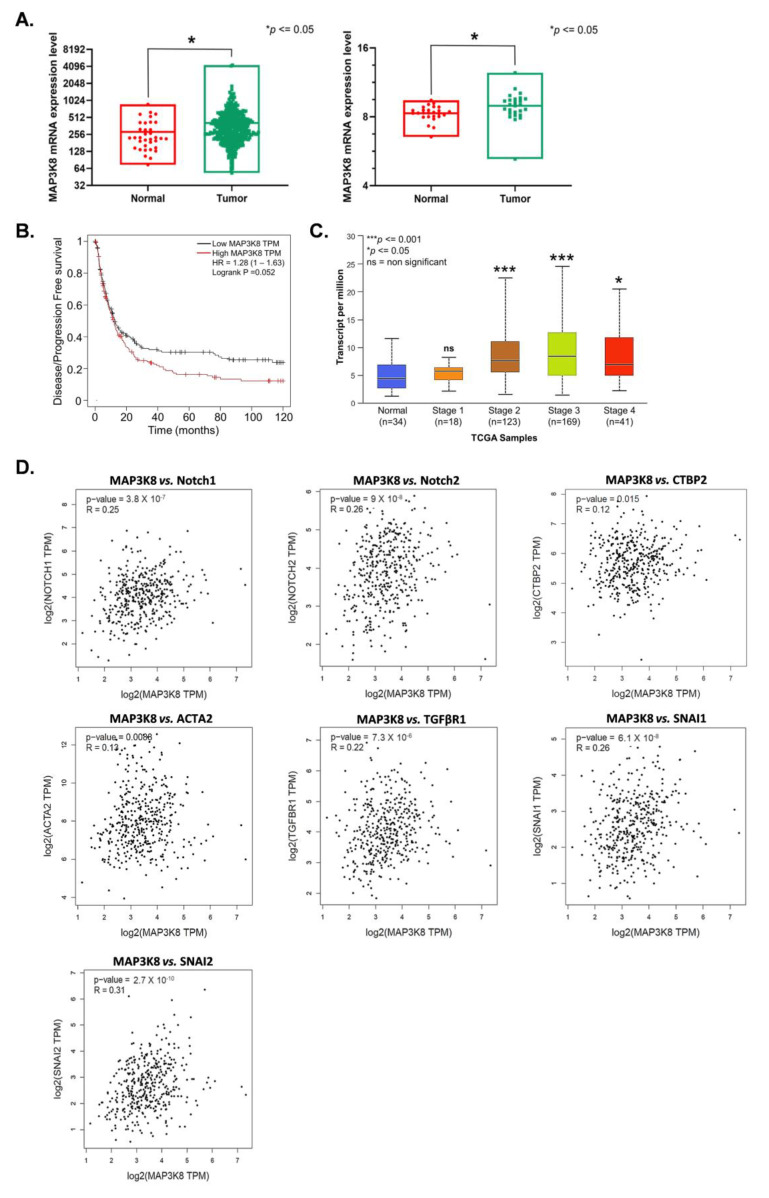
*MAP3K8* expression in STAD and its correlation with notch pathway and EMT related genes. (**A**) *MAP3K8* mRNA expression levels in GC were computed from the TCGA (left) and GSE51575 (right) databases. (**B**) Kaplan–Meier (KM) plot of *MAP3K8* expression vs. disease free progression in GC, generated using the KM plotter. The KM plot was conceived using the median expression, according to the website’s instructions. (**C**) The UALCAN web portal demonstrated the association between *MAP3K8* expression in STAD and individual cancer stages. (**D**) GEPIA-assisted Spearman’s correlation analysis to find the pair-wise gene expression correlation in STAD. Correlation coefficient (R) indicates the association between the gene pairs (*** *p* ≤ 0.001, * *p* ≤ 0.05, and ns = non-significant).

## Data Availability

The raw sequencing data reported in this paper have been deposited in the Genome Sequence Archive of the BIG Data Center at the Beijing Institute of Genomics, Chinese Academy of Science, under accession number HRA002658, accessible at https://ngdc.cncb.ac.cn/gsub/ (accessed on 11 January 2023). The authenticity of this article has been validated by uploading the key raw data onto the Research Data Deposit public platform (www.researchdata.org.cn), with the approval number as RDDB2023766851. Source data are provided with this paper.

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
