# Peer review of "Epigenetic Regulation of MAP3K8 in EBV-Associated Gastric Carcinoma"

_ijms, 2023, doi:10.3390/ijms24031964_

Round 1
Reviewer 1 Report
In the manuscript entitled “Epigenetic regulation of MAP3K8 in EBV-associated Gastric Carcinoma” the authors elucidated the role of MAP3K8 in EBV associated GC. The authors used different sophisticated techniques like HiChIP to prove the hypothesis. The topic is very interesting and of great interest in the field of EBV associated oncogenesis. The study although has lot of potentials but it lacks in the design.
1. The authors used sophisticated modern techniques like HiChIP that is normally used to detect the 3D conformation and long range interaction. But I don’t see the utility of this experiment in this context. In other words to my understanding the authors did not use the data properly. A lot could have been done with that data.
2. I don’t think the title is appropriate. The manuscript described mainly about the involvement of the MAP3K pathway and how inhibition of this pathway affects the cell growth progression and metastasis. But the title mentions about the epigenetic regulation. I don’t see a single piece of data after Figure 2 that describes about the epigenetic regulation. In the manuscript the authors illustrated the effect of the KD of the MAP3K pathway but how is that related to the epigenetic control?
3. In SNU719 cells the CRISPRi technology is used and the CRISPRi were designed for 4 different regions and significant reduction of the MAP3K8 expression was observed. But in YCCEL1 cells the change is not so obvious. Although the authors mentioned that their may be other factors that may regulate the expression, but still it raises the question about the importance of these SE in the context of MAP3K8 expression.
The concern becomes more evident when the authors showed the data with OTXO15 where the expression and viability is severely altered. That means somehow the SEs that the authors are targeting is not so important in context of the cell viability and expression of the MAP3K.
4. Also it would be more convincing if the authors can show the expression of MAP3K by WB for the CRISPRi similar to Figure 2F, I.
5. I would request the author to represent the cell viability in terms of % as compared to the untreated control and not as absorbance.
Reviewer 2 Report
The author performed target nomination through super enhancer theory, and validation their target MAP3K8 in functional assay, MTT, migration assays. Concluded that the decrease amount of MAP3K8 could limit the tumor proliferation and migration ability in cell culture models. The author suggests evidence Notch signal pathway be a mechanism of action.
1. The author probably needs to block Notch signal pathway and KD MAP3K8 and if the functional assay effect disappears, then the author can claim the mechanism of action of MAP3K8 in EBVGC is via Notch. Or Notch is necessary for MAP3K8 effects on the functional assays. At the moment, KD of MAP3K8 leads to functional assay changes, and also leads to Notch gene expression changes by RNA-seq. These two observations could be independent or associated. Further experiments are needed to determine the causal relationship. Author claims Notch signal is a necessary component of KD MAP3K8’s functional output without the necessary support. The discussion part should be clarified.
2. Biological replicated used in each graph need to be specified.
3. The calculation and statistics need more clarification, the control bar in many figures does not show variations. Either normalizer and controls are confused or some parts of the math is not correct.
Abstract
Line 21
The author claimed to be first to report epigenetic regulation of MAP3K8 in EBVGC.
It is very rare to claim first in such a broad and bold statement. The author could check https://genomemedicine.biomedcentral.com/articles/10.1186/s13073-021-00970-3 or https://pubmed.ncbi.nlm.nih.gov/27677335/ or similar, the SE studies in this report is likely referred to or reported in other large scale screening paper. I suggest the author to narrow their claim to perhaps “first to functional interrogate the mechanism of super enhancer mediated regulation of MAP3K8 in EBVGC cell lines”.
Introduction
Line 60, the function of some super enhancers in gene expression are validated using Cas9 deletion. In the 2016 paper, CLDN4 predicted super-enhancer region are confirmed using CRISPR-Cas9 deletion in OCUM-1 and SNU16 cells.
oncogenic amplification is a term used by author is perhaps need to better defined, are author referring to oncogene ampliation in somatic cells or tumorigenic/increase proliferation. How is SE activity leads to gene amplification needs to be further clarify? Leads to downregulation of DNA repair gene expression, etc.
Line 72, reviewer suggest to change the word “interesting”, to “impactful”
Results
Fig 1A cut off threshold between “normal” and “super” enhancer needs to be further justified.
Fig 1C how many biological replicates are performed during RT-qPCR.
In Fig 1, it is not clear what is the normal control? During target discovery or nomination, the disease vs control logic is perhaps usually used. The EBV-Gastric cancer cells have these SE, but control cells (non EBV-GC or normal gastric cells) do not have these SE. or the SE are always present, but EBV-Gastric cancer cells have unique 3D Enhancer-promoter loops. The evidence is not clearly presented to support that MAP3K8 super enhancer is specific to the specific indication.
Fig 2A, how many biological replicates are performed? The data is presented in a way that leads review to suspect control was misused as normalizer. The control bar does not have any error bars. What is the time frame of the RNA isolation. Later on in Fig3, the siRNA was assayed at 24hrs, was these dCas9KRAB assays also performed at 24 hrs after the 7 days lenti. The lenti is known to be unstable over passage 2 or passage 3 in certain cell lines. Is the repression by KRAB stable over passages. No Western Blot was reported at KRAB assay, but the siRNA treatment did have WB, the reviewer suggests the author justify the choice, perhaps due to the low effects of KD by KRAB.
A typical dCas9-KRAB assay studying multiple enhancers involves a mixed gRNA condition also, the author could consider that for this assay. The effect of inhibiting the super enhancer is 20% reduction of the target gene expression, with unknown effect on protein abundance. And unknown effects on MTT or migration assays. All the functional assay reported are either small molecules (not specific to MAP3K8 super enhancers) or siRNA knockdown.
Fig 2 the usage of OTX015 at specific concentration need to be justified, the author could reference https://cancerci.biomedcentral.com/articles/10.1186/s12935-022-02443-y or similar and justified the rational of 20 µM, which is on the higher end of small molecule usage in cell culture.
Fig 3B has an important difference with other MTT assay results, the NC control usually start from 0.25 and reach 2 at 120hrs. however, this NC group is very low, less than 1.5. Each this NCsiRNA has a specific effect on SNU179 cells, or the assay condition was different from other reported results.
Fig 3 bar plot continue to have no error bar, further suggestion the control samples could be misused as normalizer. Each biological replicates should have its controls. The migration assay are usually plotted as percent of cells. https://link.springer.com/article/10.1007/s00784-019-02957-2 The author could refer this or similar to consider the calculation of data.
Fig 4 The review suggest author to clarify if RNA sequencing biological replicates larger than or equals 3. Is the RNA-seq performed on NCsiRNA control also. How many and how much are the DEGs? The RNA-seq data is very valuable. It seems it is not analyzed or reported in full.
Method
Statistical test, student t-test used in prism has sub-variants. Multiple test correction is probably used in the MTT assay analysis reporting. The review suggests the author to further revise their description of the statistic used in prism.
Reviewer 3 Report
The authors research the role of MAP3K8 in EBV gastric cancer cell lines and associate with proliferation. In addition they try and translate to gastic cancer and aasociate with the NOTCH and EMT related genes. The amount of work that has been performed is impressive, the rational and conclusions are not always clear and could be explained better. What is the role of EBV? Is it on epigenetic regulation as the title implies?
1. Why is MAP3K8 so prominently part of the introduction and discussion, it seems this was the starting point and not the CHIP and HiCHIP seq trying to find super enhancer associated genes.
2. Why the comparison with GES? Is EBV important in this research or could EBV-negative GC also been used? Do you just want to compare to normal epithelial, than you will also find especially cancer specific differences. What was your starting point? EBV?
3. Why pick MAP3K8 from figure 1, not the only gene?
4. Can you explain in the text the use of the BET inhibitor, why, what was the incentive? There would probably a lot of effects of the BET inhibitor, not only MAP3K8, the effect on protein level is indeed not very high while the effect on proliferation is very good. You start out with the SE associated genes, and that fits here but then it stops.
5. Are the Boyden chamber effects specific or at a ratio of the effect of cell dead?
6. In part 2.4 from the GSEA results NOTCH is indicated, but is that all. The EMT associated genes did not show up it seems, why did you pick them?
7. The results from 2.5: this is on STAD, no EBV, please check is it EBV specific?
8. The spearman correlations are not very impressive. An R of 0 to 0.3 is no correlation and an R of 0.31 for the correlation with SNAI2 is a low correlation.
Round 2
Reviewer 1 Report
The authors responded to all the queries and the manuscript can be accepted in the current form with spell check and language editing if required.
Author Response
Thank you for your comments and approval.
Reviewer 3 Report
I don't see major improvements and am not impressed with the answers
Author Response
Thank you for your comments. We have further removed the grammatical and typographical mistakes to improve the quality of the manuscript.